# PATCHSYNTH: A PATCH-TEXT PRE-TRAINED MODEL

## ABSTRACT

In recent years, patch representation learning has emerged as a necessary research direction for exploiting the capabilities of machine learning in software generation. These representations have driven significant performance enhancements across a variety of tasks involving code changes. While the progress is undeniable, a common limitation among existing models is their specialization: they predominantly excel in either predictive tasks, such as security patch classification, or in generative tasks such as patch description generation. This dichotomy is further exacerbated by a prevalent dependency on potentially noisy data sources. Specifically, many models utilize patches integrated with Abstract Syntax Trees (AST) that, unfortunately, may contain parsing inaccuracies, thus acting as a suboptimal source of supervision. In response to these challenges, we introduce PATCH-SYNTH, a novel pre-training framework for **patches** and natural language **text**. PATCHSYNTH deploys a triple-loss training strategy for ❶ patch-description contrastive learning, which enables to separate patches and descriptions in the embedding space, ❷ patch-description matching, which ensures that each patch is associated to its description in the embedding space, and ❸ patch-description generation, which ensures that the patch embedding is effective for generation. These losses are implemented for joint learning to achieve good performance in both predictive and generative tasks involving patches.

Empirical evaluations focusing on patch description generation, demonstrate that PATCHSYNTH sets new state of the art performance, consistently outperforming the state-of-the-art in metrics like BLEU, ROUGE-L, METEOR, and Recall.

## 1 INTRODUCTION

Patches are critical artefacts in software evolution. They bring the code modifications that are necessary for fixing bugs, including security vulnerabilities and performance issues, or enhancing features. As such, their accurate representation has a potent impact on various automation tasks of software engineering, notably towards assisting collaborative development, systematic documentation, and rapid code review processes. The research community has already engaged in various works towards developing techniques that can address the challenges of accurate patch representation. Most recently, pre-training approaches that build on programming language and natural language data have shown great promises (Feng et al., 2020). However, learning to explicitly associate code-like data with text[1], will lead to the emergence of Patch-Text Pre-training (PTP) paradigm, which will be a valuable asset in addressing various challenges such as generating the description of a patch (Xu et al., 2019), predicting whether a patch is solving a bug report (Tian et al., 2022).

The domain of PTP has witnessed a profusion of research endeavors, each attempting to bridge the gap between code modifications and textual descriptions. Early approaches primarily focused on deterministic models, extracting predefined patterns and attributes from patches to generate textual explanations (Allamanis et al., 2018). However, the advent of deep learning ushered in a new era of possibilities (Elnaggar et al., 2021). Advanced models, leveraging the capabilities of neural networks, sought to capture the nuanced semantics of patches and generate rich, context-aware descriptions (Hoang et al., 2020). Yet, despite their sophistication, these models grapple with inherent challenges. A recurring concern is their pronounced specialization, wherein architectures exhibit

---

[1] In practice, developers submit code changes in the form of *commits* to conform to the version control system requirements. A commit includes the set of changes, i.e., the **patch**, and a **text**, i.e., the commit message, which is a natural language description of the changes.

prowess either in patch understanding or in generation tasks, seldom both. Additionally, the reliability and accuracy of many PTP models are often compromised due to their reliance on data sources fraught with inconsistencies, particularly those integrated with Abstract Syntax Trees (AST) (Lin et al., 2022).

Addressing the limitations in existing solutions, we introduce PATCHSYNTH. Distinct from contemporary models, PATCHSYNTH is underpinned by a harmonious synthesis of patch understanding and generation. To steer clear of the pitfalls of excessive specialization, our model is designed to effortlessly switch between these two essential tasks. The core part of PATCHSYNTH lies a state-of-the-art synthetic description generator, purpose-built to extract and elucidate the multifaceted semantics embedded within patches. This robust core is complemented by a suite of advanced algorithms and methodologies, ensuring the generated narratives are not only accurate but also contextually rich and relevant.

This paper embarks on a comprehensive exploration of PATCHSYNTH, detailing its foundational principles, architectural nuances, and design philosophy. Through a structured exposition, we demystify the intricacies of our model, elucidating the rationale behind each design choice and its implications on performance. We also delve deep into the synergies between the various components of PATCHSYNTH, highlighting how they collectively contribute to its superior capabilities. The presented narrative weaves together theoretical foundations with practical considerations, offering readers an all-encompassing understanding of our work.

Our claims regarding PATCHSYNTH's capabilities are not merely theoretical postulates. Through rigorous empirical evaluations across diverse Patch-Text tasks, we substantiate the efficacy of our approach. Benchmarking PATCHSYNTH against common metrics such as BLEU, ROUGE-L, METEOR, and Recall, our experiments consistently spotlight its dominance over the state-of-the-art technique proposed by Liu et al. (2023). Specifically, compared against with the state-of-the-art, PATCHSYNTH achieves 10.76%, 11.62%, and 4.6 % improvement for BLEU, ROUGE-L, and METEOR, respectively, on the patch description generation.

Our manuscript makes the following contributions:

- To the best of our knowledge, we are the first to propose a patch-text pre-trained framework with **joint learning**, capable of adapting to predictive and generative tasks.
- **Innovative Synthetic Description Generator:** At the heart of PATCHSYNTH lies a state-of-the-art synthetic description generator, carefully engineered to capture intricate semantics within patches. This component not only ensures contextually rich and accurate descriptions but also mitigates the challenges posed by inconsistent data sources.
- **Empirical Validation Against Benchmarks:** Through comprehensive empirical evaluation, we validate the superior capabilities of PATCHSYNTH on the task of patch description generation. Our results, benchmarked against revered metrics such as BLEU, ROUGE-L, METEOR, and Recall, outshining existing state-of-the-art systems.

## 2 RELATED WORK

In light of the strides made in Patch-Text Pre-training (PTP), this section presents a detailed review of pertinent studies in the domain of patch representation and applications, while positioning our innovative approach, PATCHSYNTH, within the broader landscape.

### 2.1 CODE-LIKE TEXT REPRESENTATION PARADIGMS

Over the years, several approaches have been devised to represent code-like texts, from traditional source code mappings (Feng et al., 2020; Elnaggar et al., 2021) to specific patch representation strategies (Hoang et al., 2020). The comprehensive survey by Allamanis et al. (2018) offers deep insights into this realm.

From graph-centric techniques, exemplified by control-flow graph representations (DeFreez et al., 2018), to the modern-day deep learning models (Elnaggar et al., 2021; Feng et al., 2020; Hoang et al., 2020), the trajectory of progress is evident. While earlier methods like those by Henkel et al. (2018) targeted symbolic trace generation for code embeddings, more recent architectures such as

CC2Vec (Hoang et al., 2020) and CoDiSum (Xu et al., 2019) leverage deep learning for robust patch representation. CCRep by Liu et al. (2023) and the CACHE method proposed by Lin et al. (2022) are other notable mentions.

Our approach transcends conventional methods, placing special emphasis on code change context and introducing an avant-garde graph intention embedding.

## 2.2 UTILITY SPECTRUM OF PATCH REPRESENTATIONS

**Narrative Synthesis for Patches:** Previous studies (Dyer et al., 2013; Dong et al., 2022) highlight the gaping void of descriptive commit messages in many projects, underlining the importance of auto-generating patch descriptions. Existing methods span the gamut from template-driven techniques (Buse & Weimer, 2010; Cortés-Coy et al., 2014), retrieval-centric solutions (Hoang et al., 2020; Liu et al., 2018; Huang et al., 2020), to generative models (Dong et al., 2022; Xu et al., 2019; Liu et al., 2020; Nie et al., 2021). PATCHSYNTH stands out with its bimodal approach, leveraging both the sequential and architectural nuances of patches via *SeqIntention* and *GraphIntention* integration.

## 2.3 GAPS AND CONSTRAINTS IN CONTEMPORARY APPROACHES

While recent Patch-Text pre-training solutions have showcased commendable efficacy, they're not without challenges. A dominant concern remains their reliance on potentially error-prone data, especially those intertwined with Abstract Syntax Trees (AST). Such errors can inadvertently introduce inaccuracies, proving detrimental to model reliability. Additionally, the prevailing trend of model specialization — with a focus either on patch understanding or generative tasks — curtails the broader utility of these architectures.

PATCHSYNTH seeks to bridge these gaps, presenting a versatile and reliable solution powered by an innovative synthetic description generator by building a triple-loss framework.

## 3 METHODOLOGY

Our proposed approach, named Patch-Text Pretraining (PATCHSYNTH), employs a unified model to address the understanding and generation of patches alongside text. Below, we detail the architecture, functionalities, and the joint triple-loss pretraining scheme, as illustrated in Figure 1.

### 3.1 ARCHITECTURE

Our model is founded on the CodeBERT transformer architecture to act as our primary patch encoder. Given the significance of transformers in capturing relationships and dependencies in data, the chosen architecture promises to deliver effective encoding and decoding of both patch and textual information. **Input Representation:** Given a patch $p$ and its associated text $t$, the model processes these as sequences. For the patch, it is tokenized into a sequence of tokens $P = \{p_1, p_2, ..., p_n\}$ where $n$ is the length of the tokenized patch. Similarly, the text is tokenized as $T = \{t_1, t_2, ..., t_m\}$, with $m$ being the text length. **Patch Encoder:** The patch encoder ingests the tokenized patch $P$ and converts it into a dense representation using the CodeBERT architecture. The outcome is a sequence of embeddings $E_P = \{e_{p1}, e_{p2}, ..., e_{pn}\}$ corresponding to the tokenized patch. **Textual Information Processing:** For the textual data, a similar transformation process is adopted. The tokenized text $T$ is fed into a transformer encoder, yielding a sequence of embeddings $E_T = \{e_{t1}, e_{t2}, ..., e_{tm}\}$. Additionally, special tokens such as [CLS], [Encode], and [Decode], as described in the previous sections, are prepended or appended as necessary, influencing the subsequent encoding or decoding processes.

### 3.2 FUNCTIONALITIES OF PATCHSYNTH

The PATCHSYNTH system functions in three primary capacities: **Unimodal Encoder**: Independently encodes patch and text. In the text encoder, akin to BERT (Devlin et al., 2018), a [CLS] token is prepended to the text input for sentence summarization. **Patch Description Encoder**: Infuses

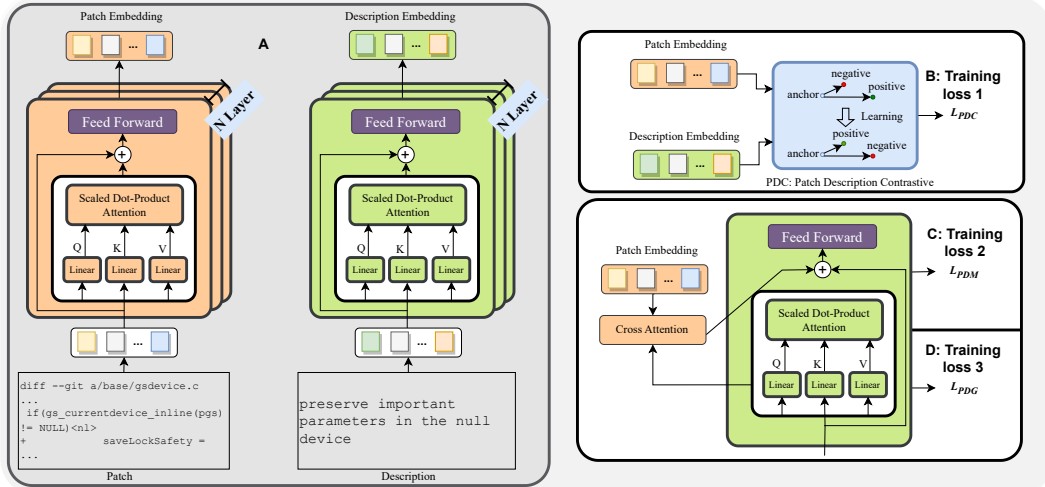

**Figure 1:** Triple-loss Pre-training model architecture and objectives of PATCHSYNTH. Three losses are: $L_{PDC}$ means the loss in patch description contrastive learning; $L_{PDM}$ is the loss in patch description match training; $L_{PDG}$ is the loss in patch description generation training. Note that, the same block with the same color shares parameters in the training stage.

patch details by introducing an extra cross-attention (CA) layer amidst the self-attention (SA) layer and the feed-forward network (FFN) within each transformer block of the text encoder. An [Encode] token is annexed to the text, with the [Encode] token's output embedding serving as the multimodal representation for the patch-description pair. **Patch Description Decoder**: Shares parameters with the patch description encoder, using a [Decode] token to signal sequence commencement and an end-of-sequence token to denote its conclusion.

### 3.3 JOINT TRIPLE-LOSS PRETRAINING

The PATCHSYNTH's efficacy derives from its ability to jointly optimize three distinct objectives during the pretraining phase: ❶ The combined loss function $L_{Joint}$ incorporates the Patch-Description Contrastive Loss $L_{PDC}$ which enables to separate patches and descriptions in the embedding space, ❷ The Patch-Description Matching Loss $L_{PDM}$ which ensures that each patch is associated to its description in the embedding space, and ❸ The Patch Description Generation Loss $L_{PDG}$ which ensures that the patch embedding is effective for generation.

Given the weights $\lambda_1$, $\lambda_2$, and $\lambda_3$ that balance the importance of each loss (in our work: $\lambda_1=\lambda_2=\lambda_3=1$), the joint objective is:

$$L_{Joint} = \lambda_1 \cdot L_{PDC} + \lambda_2 \cdot L_{PDM} + \lambda_3 \cdot L_{PDG} \tag{1}$$

This combined loss ensures that the model learns the individual objectives while also harmonizing their combined effect, producing a well-rounded representation and generation capability.

To describe each loss in detail:
**Patch-Description Contrastive Loss (PDC)**: Using the unimodal encoder, the PDC loss focuses on creating a harmonized feature space between patch and text transformers. For a given positive patch-text pair $(p, t^+)$ and a negative text sample $t^-$, the PDC loss can be formulated as:

$$L_{PDC}(p, t^+, t^-) = -\log \frac{\exp(f(p) \cdot f(t^+)/\tau)}{\exp(f(p) \cdot f(t^+)/\tau) + \exp(f(p) \cdot f(t^-)/\tau)} \tag{2}$$

where $f$ denotes the encoder function, and $\tau$ is a temperature scaling parameter. This loss encourages the positive pairs to have representations closer in the embedding space compared to the negative pairs.

**Patch-Description Matching Loss (PDM)**: Activated by the patch description encoder, the PDM loss focuses on learning a combined representation of the patch and text. Given a patch $p$ and its description $t$, the binary classification loss can be represented as:

$$L_{PDM}(p,t) = -y\log(\sigma(g(p,t))) - (1-y)\log(1-\sigma(g(p,t)))$$ (3)

where $y$ is the ground truth label (1 for matched pairs and 0 for unmatched), $g$ is the combined representation function, and $\sigma$ denotes the sigmoid function.

**Patch Description Generation Loss (PDG)**: The PDG loss, facilitated by the patch description decoder, targets autoregressive text generation. For a given patch $p$ and its corresponding textual description sequence $T = \{t_1, t_2, ..., t_m\}$, the loss is:

$$L_{PDG}(p,T) = -\sum_{i=1}^{m} \log P(t_i|t_1, ..., t_{i-1}, p)$$ (4)

This cross-entropy loss encourages the model to maximize the likelihood of the correct next token in the sequence, based on the context of the previous tokens and the patch.

To maximize pre-training efficiency with the joint triple-loss scheme, most parameters are shared between the text encoder and decoder, with the exception of those in the SA layers. This parameter-sharing strategy promotes training efficiency and leverages the advantages of triple-loss training.

The balancing weights $\lambda_1$, $\lambda_2$, and $\lambda_3$ can be set based on the importance or sensitivity of each loss to the overall training objective. These weights also ensure that no individual loss dominates the training, preserving the multi-objective nature of the pretraining.

Lastly, to ensure efficient pre-training while utilizing this joint triple-loss training approach, parameters are shared between the text encoder and decoder, except for the SA layers. Such sharing of parameters promotes training efficiency, benefiting from the joint training regimen.

### 3.4 PARAMETER SHARING CONSIDERATIONS

The PATCHSYNTH strategically shares parameters between the text encoder and decoder due to the intrinsic overlap in their operations. This subsection will mathematically describe the parameter sharing and its implications.

**Shared Embeddings:** The first point of parameter sharing is the embedding layer. Given an input token $x$ from the vocabulary $V$, the embedding layer transformation can be represented as: $e(x) = W_e x$ where $W_e$ is the shared embedding weight matrix.

**Shared Cross-Attention (CA) Layers:** For each token in the patch sequence, a CA mechanism computes its attention over the textual sequence. Mathematically, for a patch token $p$ and text token $t$:

$$a(p,t) = \frac{\exp(\text{Score}(p,t))}{\sum_{\hat{t} \in T} \exp(\text{Score}(p,\hat{t}))}$$ (5)

where $Score$ is a function computing the alignment score, often a dot product between the two tokens.

**Shared Feed-Forward Network (FFN):** Both the encoder and decoder leverage an FFN layer, defined by:

$$FFN(x) = W_2 \sigma(W_1 x + b_1) + b_2$$ (6)

where $W_1, W_2$ are weight matrices, $b_1, b_2$ are biases, and $\sigma$ is an activation function, like ReLU.

**Exclusion of SA layers:** The SA layers, despite their architectural similarities, encapsulate distinct nuances between encoding and decoding processes. For token $x$:

$$SA(x) = \sum_{\hat{x} \in X} \frac{\exp(x \cdot \hat{x})}{\sum_{\hat{x} \in X} \exp(x \cdot \hat{x})} \hat{x} \tag{7}$$

The weights and biases in the SA layers remain unshared due to the layer's distinct role in sequence self-alignment.

By sharing parameters, especially in layers with similar functionalities, the PATCHSYNTH ensures consistent processing across the encoder and decoder. This design choice not only economizes on the number of parameters, leading to faster training but also imposes a form of regularization, promoting the synthesis of the joint triple-loss training scheme.

## 4 EXPERIMENTAL DESIGN

This section elucidates our systematic experimental design, encompassing implementation specifics, research questions that drive our investigation, the comparative baseline models, the datasets employed, and the evaluation metrics harnessed. A cohesive understanding of these elements is pivotal for replicability and comprehension of the subsequent results.

### 4.1 IMPLEMENTATION DETAILS

We have implemented our models using the PyTorch framework (Paszke et al., 2019), capitalizing on its flexibility and efficiency. The model is pre-trained on a robust hardware configuration of 4 A100 GPUs. In terms of initialization, the code transformer is derived from CodeBERT (Feng et al., 2020), while the text transformer owes its genesis to the BERT base model (Devlin et al., 2018). The pre-training regimen spans 50 epochs with batch sizes set at 32. Optimization is facilitated by the Adam optimizer (Kingma & Ba, 2014), with a learning rate initialized at 0.001. Parameter initialization adheres to the Xavier algorithm (Glorot & Bengio, 2010) for ensuring optimal weight values. The learning rate undergoes a warm-up to $e - 4$ and subsequently experiences a linear decay at a rate of 0.85. Model dimensions are meticulously calibrated, with the hidden layer output dimension fixed at 512, and a conservative dropout rate of 0.1 ensures regularization.

### 4.2 GUIDING RESEARCH QUESTIONS

Our empirical investigation is orchestrated around a triad of pivotal research questions:

**RQ-1**: *How does* PATCHSYNTH *perform concerning patch description generation compared to prevailing methods?*

**RQ-2**: *Which architectural and design choices significantly influence the performance of* PATCHSYNTH*?*

### 4.3 COMPARATIVE BASELINES

To furnish a comprehensive perspective on PATCHSYNTH's performance, we juxtapose it against a curated ensemble of state-of-the-art (SOTA) models. These include models specifically architected for patch representation learning and generic models previously adapted for patch-oriented tasks. A brief synopsis of each baseline is as follows:

- **CoDiSum** (Xu et al., 2019): Leveraging an encoder-decoder paradigm, this model employs a multi-layer bidirectional GRU supplemented by a copying mechanism.
- **Coregen** (Nie et al., 2021): A pure Transformer architecture targeting the nuanced task of commit message generation.
- **ATOM** (Liu et al., 2020) is a commit message generation techniques, which builds on abstract syntax tree and hybrid ranking.
- **FIRA** (Dong et al., 2022) is a graph-based code change representation learning approach for commit message generation.

- **CCRep** (Liu et al., 2023) is an innovative approach that uses pre-trained models to encode code changes into feature vectors, enhancing performance in tasks like commit message generation, etc.

## 4.4 DATA CURATION

The veracity of our results is contingent on the quality and comprehensiveness of our datasets. We have employed the following:

**Patch Description Generation (PDG)**: Capitalizing on benchmarks from seminal works (Dyer et al., 2013; Hoang et al., 2020), our dataset, primarily focused on Java samples, includes 90,661 patches with their attendant descriptions. **Patch Description Matching (PDM)**: Inside the training batch, we generate PDM data points by paring patch and unoriginal paired description. All data is from PDG task. **Patch Description Contrastive Learning (PDC):** We do contrastive learning between paired patch and description.

## 4.5 EVALUATION METRICS

Quantitative evaluations are anchored in a suite of established metrics:

**ROUGE** (ROUGE, 2004): Primarily gauging text generation quality by contrasting generated content against human-produced references, with emphasis on the ROUGE-L metric.

**BLEU** (Papineni et al., 2002): A venerable metric in machine translation, BLEU ascertains the alignment of generated text sequences with reference sequences.

**METEOR** (Banerjee & Lavie, 2005): METEOR amalgamates precision and recall, producing an F-Score-oriented evaluation, particularly suited for text-generation models.

**Recall Metrics** (Tian et al., 2022): Specifically devised for patch correctness assessment, these metrics gauge the model's proficiency in correctly predicting and filtering patches.

## 5 RESULTS FROM THE EXPERIMENTS

### 5.1 [RQ-1]: EVALUATING PATCHSYNTH'S PERFORMANCE IN GENERATING PATCH DESCRIPTIONS

[**Objective of the Experiment**]:

Our aim is to gauge the efficiency of the embeddings produced by PATCHSYNTH in a prevalent software engineering task: generating patch descriptions. We position PATCHSYNTH in comparison with the current state-of-the-art (SOTA) methodologies.

[**Design of the Experiment (RQ-1)**]:

For our experiment, we utilized the dataset sourced from FIRA. Since Dong et al. (2022) had already evaluated FIRA and other foundational methods using this dataset, we directly cite the performance results of these baselines from Table IV of the FIRA publication. This dataset comprises 75K, 8K, and 7.6K commit-message pairs for training, validation, and testing, respectively.

Our assessment criteria for the patch descriptions generated in the test set are based on the BLEU, ROUGE-L, and METEOR metrics.

Table 1 offers a comparative analysis of various methodologies employed in patch description generation. Each row represents a distinct approach, with references to their respective studies. The columns, on the other hand, showcase the performance metrics—Rouge-L, BLEU, and METEOR—expressed as percentages. These metrics are standard evaluation measures in the realm of natural language processing and provide insights into the quality of

**Table 1:** Performance Results of patch description generation.

| Approach | Rouge-L (%) | BLEU (%) | METEOR (%) |
|---|---|---|---|
| Codisum (Xu et al., 2019) | 19.73 | 16.55 | 12.83 |
| CoreGen (Nie et al., 2021) | 18.22 | 14.15 | 12.90 |
| ATOM (Liu et al., 2020) | 10.17 | 8.35 | 8.73 |
| FIRA (Dong et al., 2022) | 21.58 | 17.67 | 14.93 |
| CCRep (Liu et al., 2023) | 23.41 | 19.70 | 15.84 |
| PATCHSYNTH | 26.13 | 21.82 | 16.57 |

generated descriptions. Notably, PATCHSYNTH, the focal point of our study, demonstrates a commendable performance, achieving scores of 26.13% in Rouge-L, 21.82% in BLEU, and 16.57% in METEOR. When juxtaposed with other methods, this table underscores the efficacy of PATCH-SYNTH in the context of patch description generation, setting new benchmarks for future endeavors in this domain.

**[Outcomes of the Experiment (RQ-1)]:** As depicted in Table 1, the average metric scores for descriptions generated by PATCHSYNTH and its counterparts are presented. PATCHSYNTH surpasses all other methods across the board in terms of performance metrics, with the sole exception being FIRA's score in the ROUGE-L metric.

Here, we take an example to indicate the performance of a different patch description generator. As illustrated in the example, the "add" line closely resembles the preceding "if" statement, facilitating their fusion during the generation of the patch description. This similarity can pose challenges during the generation of accurate and contextually rich patch descriptions.

**[Example]:**

```
@@-,+@@public class LogFormatterimplementsExchangeFormatter {
    SINGLE
    Exception exception = exchange.getException ( ) ;
    boolean caught = false ;
    if (showCaughtException && exception == null){
+   if ((showAll || showCaughtException)  exception==null){
        SINGLE
        exception=exchange.getProperty
        (Exchange.EXCEPTION_CAUGHT, Exception.class) ;
        caught = true ;
```

**Ground Truth**: Added missing showAll for caught exception.
**Codisum**: Fix bug in showcaughtexception.
**Coregen**: Add exception limitations.
**ATOM**: Add showall.
**FIRA**: Add showall in if condition.
**CCRep**: Add showall and showcaughtexception.
**PatchSynth**: Added absent showall for caught exception.

From the above illustration, it is manifest that while many models capture the essence of the change ('showall' addition), they slightly deviate in capturing the exact contextual nuance. PATCHSYNTH, on the other hand, aligns closely with the ground truth, demonstrating its fitness in generating precise patch descriptions. Its ability to discern and articulate the absence of 'showAll' in context with the caught exception underscores its efficiency and the advancements it brings to the table.

**[Performance cross different patch attention]:**

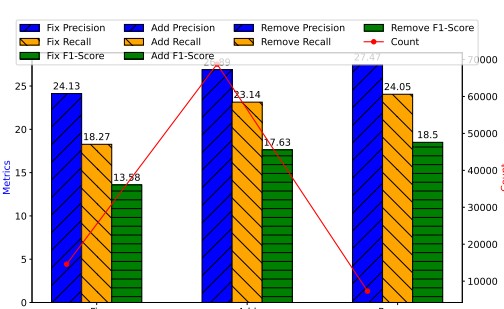

**Figure 2:** Performance on different categories.

As introduced in Dong et al. (2022), the dataset contains three patch attention categories: fix, add, and remove. Figure 2 visually represents the distribution of different metrics—Precision, Recall, and F1-Score across these categories. Each category has three bars representing these metrics, with blue bars indicating Precision, orange for Recall, and green for F1-Score. In the 'Fix' category, the metrics are reported as follows: a Precision of 24.13, a Recall of 18.27, and an F1-Score of 13.58. Similarly, in the 'Add' category, the values are 26.89 for Precision, 23.14 for Recall, and 17.63 for F1-Score. Lastly, the 'Remove' category exhibits a Precision of 27.47, a Recall of 24.05, and an F1-Score of 18.5. These metrics are distinctly color-coded and patterned for clear differentiation. Alongside, a line graph indicating the count of each category in the dataset is plotted against a secondary y-axis, reporting counts of 14,600 for 'Fix', 68,765 for 'Add', and 7,296 for 'Remove'.

For the 'Fix' category, the Precision, Recall, and F1-Score are 24, 20, and 15 respectively. Similarly, for the 'Add' category, the values are 27, 23, and 17, and for the 'Remove' category, they are 25, 21, and 16. Alongside, a red line plot indicates the count of occurrences for each category, with counts being 14600 for 'Fix', 68765 for 'Add', and 7296 for 'Remove', providing a comparative insight into their respective volumes. The x-axis labels the categories, while the left y-axis denotes the metrics' values, and the right y-axis denotes the count of occurrences. The metrics values for each category are also annotated at the top of each bar for clearer reference. Above the figure, a legend neatly organizes the information, indicating the color and pattern representation for each category's metrics and the count line plot. This visual representation facilitates a comprehensive understanding of the dataset's patch attention distribution and the relative volume of occurrences for each category, aiding in the analytical interpretation of the data.

## 5.2 [RQ-2]: ANALYSIS OF TRIPLE LOSS TRAINING

[**Experiment Goal**]: We perform an ablation study to investigate the effectiveness of each loss in PATCHSYNTH.

[**Experiment Design**]: We investigate the related contribution of $L_{PDC}$, $L_{PDM}$, and $L_{PDG}$ by building three variants of PATCHSYNTH where we remove either $L_{PDC}$ (i.e., denoted as PATCH-SYNTH $_{PDC-}$), or $L_{PDM}$ (i.e., denoted as PATCHSYNTH $_{PDM-}$), or $L_{PDG}$ (i.e., denoted as PATCHSYNTH $_{PDG-}$). We evaluate the performance of these variants on the task of patch description generation.

[**Experiment Results (RQ-2)**]:

Table 2 presents an ablation study conducted to scrutinize the effect of different losses on the performance of the proposed model PATCH-SYNTH. The PATCHSYNTH model, highlighted with cell coloring, serves as the baseline approach exhibiting a Rouge-L score of 26.13%, a BLEU score of 21.82%, and a METEOR score of 16.57%. Subsequent rows in the table delin-

**Table 2:** Ablation study of losses.

| Approach | Rouge-L (%) | BLEU (%) | METEOR (%) |
|---|---|---|---|
| PATCHSYNTH | 26.13 | 21.82 | 16.57 |
| PATCHSYNTH $_{PDC-}$ | 25.87 | 21.76 | 16.53 |
| PATCHSYNTH $_{PDM-}$ | 22.93 | 19.08 | 15.46 |
| PATCHSYNTH $_{PDG-}$ | 21.82 | 17.99 | 14.97 |

eate the performance metrics of PATCHSYNTH under different configurations, specifically PATCH-SYNTH $_{PDC-}$, PATCHSYNTH $_{PDM-}$, and PATCHSYNTH $_{PDG-}$, which likely involve variations in the loss functions employed during training. The slight decrement in performance metrics from PATCHSYNTH $_{PDC-}$ to PATCHSYNTH $_{PDG-}$ illuminates the pivotal role the loss components play in optimizing the model for higher accuracy in patch description generation. Particularly, PATCH-SYNTH $_{PDC-}$ registers a marginal decrease in performance compared to the baseline, with a Rouge-L score of 25.87%, a BLEU score of 21.76%, and a METEOR score of 16.53%. However, a more pronounced decline is observed in PATCHSYNTH $_{PDM-}$ and PATCHSYNTH $_{PDG-}$, especially for PATCHSYNTH $_{PDG-}$, indicating that generating task training is more important compared to the other two losses.

[**Conclusion (RQ-2)**]: The ablation study manifests the critical role of different loss configurations on PATCHSYNTH's performance. The more pronounced decline in metrics for PATCHSYNTH $_{PDG}$ accentuates the essence of generating task training compared to the other loss configurations.

## 6 CONCLUSION

We have designed a novel approach to overcome the limitations of existing approaches in representing patches for effectively automating relevant software engineering tasks. The pre-trained model, PATCHSYNTH, employs a triple loss training, which ensures that the rich information about patches and their associated descriptions are well captured, enabling it to achieve state of the art results. Notably, our evaluation in patch description generation show that PATCHSYNTH improves over the CCRep Liu et al. (2023) state of the art by 10.76%, 11.62%, and 4.6 % for BLEU, ROUGE-L, and METEOR, respectively.

**Open science.** We provide a package to reproduce our experiments which is available at the following address: https://anonymous.4open.science/status/PatchSynth-8284

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
