# OpenReview forum: "PatchSynth: a Patch-Text Pre-trained Model"
_ICLR.cc/2024/Conference — ICLR 2024 Conference Withdrawn Submission_

### Official Review · Reviewer_xxEb · 2023-10-26

**Soundness:** 4 excellent
**Presentation:** 3 good
**Contribution:** 4 excellent
**Rating:** 8
**Confidence:** 5

**Summary:**

The paper introduces a novel model, PatchSynth, in the Patch-Text Pre-training (PTP) domain, aiming to improve software patch representation and description generation. Through a blend of patch understanding and generation, PatchSynth	 addresses the limitations of prior models. Empirical evaluations reveal its superior performance in patch description generation, with an ablation study further underscoring the importance of generating task training.

**Strengths:**

Novelty and Importance: The work is first to propose a unimodel for patch-text understanding and related tasks. And the topic is very important in this domain.

	Melds patch understanding and generation, addressing prior models' specialization limitations.

	The work provides a good representation and good results

**Weaknesses:**

Unclear adaptability across diverse programming languages or coding standards.

**Questions:**

What are the considerations for deploying PatchSynth in real-world software development environments, and what infrastructure would be required for efficient and secure operation?

---

### Official Review · Reviewer_4kdr · 2023-10-26

**Soundness:** 4 excellent
**Presentation:** 4 excellent
**Contribution:** 4 excellent
**Rating:** 10
**Confidence:** 5

**Summary:**

The paper discusses a new model, PatchSynth, in the domain of Patch-Text Pre-training (PTP) which aids in accurate patch representation for software evolution tasks like bug fixing and feature enhancement. PatchSynth is designed to balance patch understanding and generation, overcoming limitations of previous models. It outperforms existing models in patch description generation, as shown in experiments using standard evaluation metrics. An ablation study further reveals the importance of generating task training in improving PatchSynth performance.

**Strengths:**

Novelty:
The novelty of PatchSynth lies in its harmonious synthesis of patch understanding and generation, coupled with an advanced synthetic description generator. This innovative approach addresses the historical challenges of accurate patch representation and description generation, marking a significant stride in the PTP paradigm.
Importance:
The topic is of paramount importance as it addresses a critical need in software engineering for accurate patch representation and description, which are pivotal for collaborative development, systematic documentation, and rapid code review processes. By advancing the PTP paradigm, PatchSynth not only contributes to the academic discourse but also holds promise for practical applications in software development workflows.
The work achieves promising results.

**Weaknesses:**

The paper doesn't elucidate how PatchSynth adapts to varying programming languages or codebases with differing coding standards and structures. This lack of demonstrated adaptability could limit its applicability across diverse software projects, potentially requiring additional tuning or re-training to maintain accuracy and effectiveness in different environments.

**Questions:**

1. Given the advancements in PatchSynth for patch-text understanding and generation, how well does the model perform in a transfer learning scenario? Can PatchSynth be fine-tuned or adapted effectively to related tasks in software engineering or different programming languages?
2. Are there considerations or plans for deploying PatchSynth in real-world software development environments? How would the integration look like, and what kind of support or infrastructure would be required to ensure the model operates efficiently and securely in a production setting?

---

### Official Review · Reviewer_H6rR · 2023-10-29

**Soundness:** 1 poor
**Presentation:** 1 poor
**Contribution:** 2 fair
**Rating:** 3
**Confidence:** 4

**Summary:**

This paper tackles the problem of code patch representation learning -- how to represent edits on code to support downstream tasks like commit message generation, patch correctness assessment, etc.

This paper proposes a pretraining framework, with triplet losses on text-code contrastive loss, text-code matching loss and text generation loss based on code patch. The pretraining data includes 90K pairs of code change and synthesized commit messages.

On downstream task of commit message generation upon FIRA[1] dataset, PatchSynth showed performance gains over public & self-ablation baselines.

[1] Jinhao Dong, Yiling Lou, Qihao Zhu, Zeyu Sun, Zhilin Li, Wenjie Zhang, and Dan Hao. Fira: Fine-grained graph-based code change representation for automated commit message generation. 2022.

**Strengths:**

1. Unlike from previous approaches(CCRep[1], Cache[2]) where code change is encoded with two streams (code-before-change, code-after-change), this work encodes code change(patch) with a standard transformer on a single patch file (like git commit diff). This is inline with the general trend in LLMs community that ultimately LLMs should be able to understand and capture internal structure without explicitly modelling it.
2. This paper applies representation pretraining with triplet losses -- which is quite known in multimodal pretraining domain (BLIP[3], BLIP-2[4], etc) -- to code patch representation. It empirically showed that such pretraining is helpful for downstream task of commit message generation.


[1] Zhongxin Liu, Zhijie Tang, Xin Xia, and Xiaohu Yang. Ccrep: Learning code change representations via pre-trained code model and query back. In 45th IEEE/ACM International Conference on Software Engineering, ICSE 2023, Melbourne, Australia, May 14-20, 2023, pp. 17–29. IEEE, 2023. doi: 10.1109/ICSE48619.2023.00014. URL https://doi.org/10.1109/ICSE48619. 2023.00014.

[2] Bo Lin, Shangwen Wang, Ming Wen, and Xiaoguang Mao. Context-aware code change embedding for better patch correctness assessment. ACM Transactions on Software Engineering and Methodology (TOSEM), 31(3):1–29, 2022.

[3] Junnan Li, Dongxu Li, Caiming Xiong, & Steven C. H. Hoi (2022). BLIP: Bootstrapping Language-Image Pre-training for Unified Vision-Language Understanding and Generation. In ICML (pp. 12888–12900). PMLR.

[4] Junnan Li, Dongxu Li, Silvio Savarese, & Steven C. H. Hoi (2023). BLIP-2: Bootstrapping Language-Image Pre-training with Frozen Image Encoders and Large Language Models. In ICML (pp. 19730–19742). PMLR.

**Weaknesses:**

I have multiple major concerns on the paper based on its current form. The most concerning issues are:

1. The paper claims "The core part of PATCHSYNTH lies a state-of-the-art synthetic description generator" in multiple places (3rd paragraph of Introduction, 2nd contribution in last part of Introduction, Section 2.3). However, there is no details on this synthetic description generator. The only mention is Section 4.4 with just one line "Capitalizing on benchmarks from seminal works[1,2], our dataset, primarily focused on Java samples includes 90,661 patches with their **attendant** descriptions"

    i. Are these **attendant** descriptions generated by the author but simply taken from [1,2]? If the latter, then claiming such synthetic description generation as a key feature in this paper is highly problematic.

2. The task of representation learning of code patch, defaultly assigns 1 vector for a code patch (w/ 1 or more edits), which is the case for all previous works including CC2Vec[3], CCRep[4], Cache[5]. However, PATCHSYNTH seems to encode the code patch to a sequence of vectors (Figure 1).

    i. Such change needs explicit explanation and justification which authors have failed to deliver.

3. Missing LLM baselines: with code patch being encoded with a sequence of vectors, the authors should compare with code-aware LLMs like Code-llama, or WizardCoder, as they also encode code patch to a sequence of vectors.

    i. As the recent code-aware LLMs have shown great abilities in general instruction following in coding-related tasks, a very timely baseline would be applying code-aware LLMs to the downstream task of commit message generation, with few-shot prompting or finetuning.

    ii. A comparison of PATCHSYNTH vs code-aware LLMs would very helpful for the community to understand the edge and relevance of the proposed method in LLM era, which the authors have failed to deliver.

4. Only 1 downstream task evaluated: The authors claimed that the method is designed both for generative and discriminative tasks. However, the empirical experiments were only conducted on commit message generation. As the encoding changed from one vector to a sequence of vectors, it's important to show how can such encoding can be adapted to tackle retrieval or classification tasks. Also, to claim it as a pretrain model, the authors need to evaluate on multiple downstream datasets.

5. Fairness in comparison:

    i. Is PATCHSYNTH firstly pretrained on 90K and then finetuned on 75K data of FIRA? If so, it's not so fair to compare PATCHSYNTH with CCRep and FIRA methods, as they are not trained on 90K pretraining data. For example, Is it possible to also pretrain CCRep with 90K data?

    ii. As mentioned in point 2, CCRep has a more compact encoding of 1 vector while PATCHSYNTH encodes to a sequence of vectors. It is thus not fair to compare without explicitly mentioning such differences.

6. Concerns on Pretraining:

    i. details of creating negative pairs: one common technique in contrastive training is hard-negative-mining. However, the authors didn't disclose how they create negative pairs

    ii. For vision-language representation learning, a large batch size (>= 512) and a large pool to select negative examples have been shown to be necessary. This paper mentions the batch size of 32, which seems pretty small. I will need more verification on ablation of 1) batch size, 2) negative example selection and 3) pretraining metrics to be convinced that such setting is adequate for code-text representation pretraining.

7. Writing & formatting issues

    i. On page 8, the chart of Figure 2 is partially blocked by its top legend

    ii. On page 8, the paragraph for [Performance cross different patch attention] is repetitive: it repeats twice in introducing the numerical performance. Besides, I don't think it's a good idea to verbosely list down all numbers when they are clearly seen in Figure 2.

    iii. In Section 2.1, there's no mention on recent code aware LLMs like Code-LLaMA.

    iv. In Section 2.3, there's no citation to any work. Besides, CCRep[4] doesn't have the gap mentioned in Section 2.3 as it can both do discriminative and generative tasks and it doesn't reply on AST information. So an explicit comparison to CCRep in Related Work should be present.

[1] Robert Dyer, Hoan Anh Nguyen, Hridesh Rajan, and Tien N Nguyen. Boa: A language and infrastructure for analyzing ultra-large-scale software repositories. In 2013 35th International Conference on Software Engineering (ICSE), pp. 422–431. IEEE, 2013.

[2] Thong Hoang, Hong Jin Kang, David Lo, and Julia Lawall. Cc2vec: Distributed representations of code changes. In Proceedings of the ACM/IEEE 42nd International Conference on Software Engineering, pp. 518–529, 2020.

[3] Thong Hoang, Hong Jin Kang, David Lo, and Julia Lawall. Cc2vec: Distributed representations of code changes. In Proceedings of the ACM/IEEE 42nd International Conference on Software Engineering, pp. 518–529, 2020.

[4] Zhongxin Liu, Zhijie Tang, Xin Xia, and Xiaohu Yang. Ccrep: Learning code change representations via pre-trained code model and query back. In 45th IEEE/ACM International Conference on Software Engineering, ICSE 2023, Melbourne, Australia, May 14-20, 2023, pp. 17–29. IEEE, 2023. doi: 10.1109/ICSE48619.2023.00014. URL https://doi.org/10.1109/ICSE48619. 2023.00014.

[5] Bo Lin, Shangwen Wang, Ming Wen, and Xiaoguang Mao. Context-aware code change embedding for better patch correctness assessment. ACM Transactions on Software Engineering and Methodology (TOSEM), 31(3):1–29, 2022.

**Questions:**

1. Why did you use CodeBERT to initiate both text encoder and decoder? For example, did you consider encoder-decoder model like Code-T5?

2. In Experiment Setup, you mentioned "Model dimensions are meticulously calibrated". May I know how are the hyper-parameters searched? Are you using the downstream task performance or some pretraining metrics?

---

### Official Review · Reviewer_639w · 2023-10-31

**Soundness:** 2 fair
**Presentation:** 1 poor
**Contribution:** 1 poor
**Rating:** 3
**Confidence:** 5

**Summary:**

Programs are frequently modified in commits, with the changes represented as program patches and often described with natural language text. This work proposes to finetune a pair of BERT encoders on the combination of such patches and their descriptions, introducing a three-fold loss. The resulting model is evaluated on a patch description generation task, where it outperforms recent baselines.

**Strengths:**

This work focuses on an established and reasonably important task in software engineering, patch representation and description (or, commit message) generation. It uses a fairly conventional encoder-decoder architecture with additional loss terms for this. Its main contribution lies in the combination of these components, which evidently yields improved performance compared to prior work.

The results show improvements in the order of 5-10% (relative) compared to baselines, which, while still yielding relative low BLEU scores (~21%), might benefit the commit message generation work and tools with a stronger baseline. The methodology, including the architecture and loss terms, was relatively easy to follow.

**Weaknesses:**

The technical contribution is very limited. The work connects two pretrained encoders (one for text and one for code) with cross-attention. Its main aim is to generate patch descriptions from the patch. It introduces two additional (but not novel) loss terms that provide a form of embedding alignment, both based on a contrastive loss. One of these appears to provide no significant benefit (DPC, Tab. 2). The setup is evaluated on a fairly small batch of mainly Java samples that appear to consist of a single patch with their associated commit message. These type of Github-scraped datasets tend to suffer from many low-quality descriptions that make it hard to meaningfully train and evaluate models and the set used in this work is no exception: the highest reported BLEU score is just 21.82%. The results corresponding to Fig. 2 show that precision/recall is about even across use-cases. As such, the work offers a useful, fairly off-the-shelf baseline for further experiments in this domain, but does not provide significant new insights or theoretical contributions.

This aside, the writing suffers from a range of problems. A number of claims about prior work are poorly motivated and several methodological details are poorly described. I list these below. More generally, the paper was quite hard to read. Many sentences include an unusual adjective or phrase that often feels overly subjective and out of place. Some examples from the first few pages: "a profusion of research endeavors", "pronounced specification", "exhibit prowess", "offering an all-encompassing understanding", "not merely theoretical postulates", "Our approach transcends conventional methods", "avant-garde graph intention embedding". It would greatly benefit the work to normalize the language, both reducing the use of highly subjective statements and replacing rare words with more commonly used synonyms.

A number of issues:

- P1: "will lead to the emergence of.." seems wrong. As noted shortly afterwards, Patch-Text (pre)training is already the subject of multiple studies. If the intent is to forecast that this particular paper will produce a new paradigm, I would strongly recommend removing this sentence.
- P2: "seldom both" -- does this imply that it is sometimes studied jointly? If so, please provide citations.
- P2: "fraught with inconsistencies, particularly those integrated with Abstract Syntax Trees" -- it is not at all clear what this means. Why are ASTs more likely to be/lead to inconsistencies? The mapping of code to ASTs is unambiguous.
- Sec 3.1: the reference to a "previous section" seems wrong; these terms were not introduced before this point.
- Sec 4.1: wrong notation in "e - 4". As written, this subtracts 4 from e.
- Sec 4.1: how exactly where the dimensions "meticulously calibrated"? The two hyper-parameters mentioned next are standard. Were hyper-parameter sweeps conducted? Please share the results of those if so.
- Sec 4.4: does this mean you combined the dataset of two prior papers, or used the same as theirs? If so, "our dataset" is wrong. If not, please elaborate on the process by which the dataset was constructed.
- Sec. 5.1: the text under "Outcomes" says that FIRA outperforms PatchSynth on ROUGE-L (Tab. 1), which it does not.
- P8: the example here seems wrong on several counts. The patch should delete the previous if-statement. The newly added line is missing "&&". The word "SINGLE" appears a few times in places where it doesn't seem to be grammatically correct for it to do so. Perhaps this is due to the odd line wrapping and indentation?
- Fig. 2: the count values should not be connected with a line; this data is not sequentially related.

**Questions:**

Please discuss whether this work offers a concrete novel technical contribution or whether it should be mainly read as setting a new baseline for patch description based on existing methods. Consider the limitations noted above: one of the loss terms does not seem to have much impact, none of the loss terms are not novel, nor is tuning a cross-attention layer between pretrained models.

Please clarify some of the methodogical questions raised above, including where the dataset came from, whether any further processing was done, and if/how the hyper-parameters were tuned.

---

### Official Review · Reviewer_9sGD · 2023-11-07

**Soundness:** 2 fair
**Presentation:** 1 poor
**Contribution:** 1 poor
**Rating:** 1
**Confidence:** 5

**Summary:**

Main contributions of the paper are:

- Proposes PATCHSYNTH, a novel pre-training framework for jointly learning patch and text representations. This allows the model to perform well on both patch understanding and generation tasks.
- Implements an innovative synthetic description generator to capture semantics within patches. This aims to mitigate issues with inconsistent data sources like error-prone ASTs.
- Uses a triple loss training strategy with losses for contrastive learning, matching, and generation. The joint training aims to harmonize the losses.
- Sets new state-of-the-art results on patch description generation, outperforming existing methods on metrics like BLEU, ROUGE-L, and METEOR.

**Strengths:**

This paper addresses the patch generation task, which is typically a difficult task in the software engineering domain. Adapting the triplet loss into the pretraining stage is somehow new.

**Weaknesses:**

The first thing I notice is that the writing is very bad and uses unnatural words, such as "Distinct from contemporary models, PATCHSYNTH is underpinned by a harmonious synthesis of patch understanding and generation. To steer clear of the pitfalls of excessive specialization, our model is designed to effortlessly switch between these two essential tasks". This looks very similar to an AI assistant tool like ChatGPT generated. Can the authors confirm that the majority of the writing was generated by AI tools?

The related section lacks a lot of related work to patch generation tasks, such as commit message generation [1, 7], code summarization [4,5,6], and patch assessment [2,3]. It appears that the author did not conduct a thorough literature review before working on this topic.

Can the authors highlight how the triplet loss contributes to the novelty of the paper?

The evaluation metrics are unclear; why are such metrics used for this task? Furthermore, the baselines used are weak and not carefully chosen. PatchSync should be compared to recent Code Large Language Models such as GPT 3.5, GPT-4, CodeGen [9], StarCoder [8], and others. I believe that simple prompting on these models can easily solve this task without using PATCHSYNTH. Finally, the benchmark datasets are old, and the purpose is not well explained due to poor writing.

Overall, I believe that this paper is poorly written, lacks a novel contribution, and the literature is poorly performed. The experiments aren't much better.


[1] Context-aware retrieval-based deep commit message generation, https://ink.library.smu.edu.sg/cgi/viewcontent.cgi?article=7779&context=sis_research

[2] Invalidator: Automated Patch Correctness Assessment Via Semantic and Syntactic Reasoning, https://ieeexplore.ieee.org/abstract/document/10066209

[3] Zero-Shot Automatic Patch Correctness Assessment, https://arxiv.org/abs/2303.00202

[4] Deep code comment generation, https://ink.library.smu.edu.sg/cgi/viewcontent.cgi?article=5295&context=sis_research

[5] Just-in-time obsolete comment detection and update, https://ink.library.smu.edu.sg/cgi/viewcontent.cgi?article=8772&context=sis_research

[6] Retrieve and refine: exemplar-based neural comment generation, https://arxiv.org/pdf/2010.04459.pdf

[7] Jointly Learning to Repair Code and Generate Commit Message, https://arxiv.org/abs/2109.12296

[8] StarCoder: may the source be with you!, https://arxiv.org/abs/2305.06161

[9] CodeGen2: Lessons for Training LLMs on Programming and Natural Languages, https://arxiv.org/abs/2305.02309

**Questions:**

Why the evaluation metrics are used for this task?

Can you provide comparison with Code Large Language Models?

Why the triplet loss is novel for this task?